# Psycholinguistics: Analysis of Knowledge Domains on Children’s Language Acquisition, Production, Comprehension, and Dissolution

**DOI:** 10.3390/children9101471

**Published:** 2022-09-26

**Authors:** Ahmed Alduais, Hind Alfadda, Dareen Baraja’a, Silvia Allegretta

**Affiliations:** 1Department of Human Sciences, University of Verona, 37129 Verona, Italy; 2Department of Curriculum and Instruction, King Saud University, Riyadh 11362, Saudi Arabia; 3Department of English Language and Literature, King Saud University, Riyadh 11362, Saudi Arabia; 4Department of General Psychology, University of Padua, 35122 Padova, Italy

**Keywords:** psycholinguistics, language acquisition, language production, language comprehension, language dissolution, scientometric review

## Abstract

This paper utilised bibliometric and scientometric indicators to assess the current state of research in psycholinguistics. A total of 32,586 documents in psycholinguistics were included from *Scopus*, *WOS*, and *Lens* between 1946 and 2022. The collected data were analysed using CiteSpace 5.8.R3 and VOSviewer 1.6.18. The results included tabulation, visualisation, and mapping for the past, present, and future directions of the field of psycholinguistics. We identified key authors, works, journals, and concepts in the existing evidence concerning (children’s) language acquisition, production, comprehension, and dissolution. The study contributes to the systematic study of existing scholarship in the field of psycholinguistics by documenting the progress of the field and informing relevant researchers about the current state of the field of psycholinguistics. Having grouped the 32,586 documents in psycholinguistics, 12 clusters were identified. These include (1) examining individual difference in affective norm and familiarity account; (2) examining refractory effect in the role of Broca’s area in sentence processing; (3) using eye movement to study bilingual language control and familiarity account; (4) exploring familiarity account through relative clauses; (5) the study of formulaic language and language persistence; (6) examining affective norm and sub-lexical effect in Spanish words; (7) examining lexical persistence in multiplex lexical networks; (8) the study of persistence through cortical dynamics; (9) the study of context effect in language learning and language processing; (10) the study of neurophysiological correlates in semantic context integration; (11) examining persistence as an acquisition norm through naming latencies; and (12) following a cross-linguistic perspective to study aphasic speakers.

## 1. Introduction

There is no doubt that children play a crucial role in the continuity of the human species. From the moment they are born, they go through several developmental stages. Language is one of these developmental stages in which they progress from acquisition to production to comprehension, and then, during any of these stages, they may experience language dissolution. Beginning with cooing, babbling, early words, and early grammar, the children will progress step by step to become able to communicate and use language appropriately. Through conceptualization, formulation, articulation, and self-monitoring, children can correctly produce language. Furthermore, they will be able to comprehend sounds, words, sentences, texts, and language beyond the words or intended meanings of the texts. In the following section, we will introduce the history of psycholinguistics; discuss its scope, including its definition(s) and scientific contributions; and conclude with the main purpose of this study.

### 1.1. The Rise of Psycholinguistics

Reviewing literature related to the history of psycholinguistics shows that many psycholinguists linked the birth of this interdisciplinary science to the Chomskyan cognitive revolution during the late 1950s and 1960s [1,2]. Interests in the mind and language, however, began much earlier. For the sake of reviewing contributions to the emergence and development of psycholinguistics, they will be selectively highlighted in this section.

Altmann attributed the earliest contributions to psycholinguistics to the ancient Egyptians, who were the first to write about language and the brain [1]. In around 1700 B.C., they mentioned in a catalogue of the effects of head injury, known now as the Edwin Smith Surgical Papyrus, the first documented case of aphasia. He added that the earliest to write about language was probably the Greek philosopher Plato (427–347 B.C.), whose writings had a great impact on the philosophy of language [1].

The establishment of this science resulted from the contact and integration between linguistics and psychology, which took about two centuries of historical development in the study of language use in mind, brain, and behaviour [3,4]. Though studying the relation between mind and language has attracted scholars for ages, the empirical roots of psycholinguistics, according to Levelt, date back as the end of the eighteenth century. These roots merged about 100 years later, and psycholinguistics became an established discipline. Then, it turned into a flourished field of study within the first half of the twentieth century. It is worth noting that by the nineteenth century, psycholinguistics was called the “psychology of language”. The term “psycholinguistics” was initiated in 1936 by American psychologist Jacob Kantor in his book *An Objective Psychology of Grammar*, but it became popular in 1946 when Kantor’s student, Nicholas Pronko, authored his article “Language and psycholinguistics: A review”. Finally, psycholinguistics developed into an academic discipline as a result of a seminar at Cornell University in 1951 [2,3,4].

Following Levelt, psycholinguistics has four empirical roots [3]. The first one arose from the search for the origins of language, and it was motivated by the discovery of the Indo-European language family. During the end of the eighteenth century, the movement from the Enlightenment to Romanticism and the emergence of historical-comparative linguistics triggered the idea of the natural origins of language and speech. Romanticism adopted a naturalistic and holistic view to the examination of both nature and mind. They attributed language emergence to natural causes rather than being a divine gift or some form of deliberate social contract. Further, the newly discovered languages, namely Asian, African, and, particularly, American Indian, increased the attraction towards the comparison. Johann Gottfried Herder and Dietrich Tiedemann are good representatives of those who wrote about language origins in 1772, yet they were not the only ones interested in the language origins during that time. There were also Jean-Jacques Rousseau, James Burnett, Lord Monboddo, and Wolfgang von Kempelen, who was the first to construct a speaking machine. The study of language in the brain was considered the second root that also appeared in the latter decades of the eighteenth century. The pioneer in this regard was Franz Gall, who, assumed that “language function was localized in the anterior parts of the brain” [1] (p. 259). Then, as Levelt, 2013, described, psycholinguistics became an advanced science during the second half of the nineteenth century, mainly after the discoveries of Broca and Wernicke [3]. Moving to the empirical study of how children acquire language, which was the third root, the study of child language was motivated by the publication of Rousseau’s *Émile* in 1762, in which he mentioned his own observations on children’s language and encouraged teachers to carefully observe the language of their students. Then, after the publication of the biographical developmental notes by Darwin in 1877, it moved to be the subject of systematic empirical study [5]. Finally, the fourth root was the experimental and speech error approaches to the language processing of normal adults. In 1865, a new research paradigm in experimental psycholinguistics called “mental chronometry”, i.e., the measurement of reaction time, brought by Franciscus Donders, who discovered and manipulated “mental processing speed”. In 1879, Wilhelm Wundt founded the first psychological laboratory in Leipzig Germany and applied this paradigm. Then, by the late 1890s, Rudolf Meringer led the modern analysis of spontaneously produced speech errors [3].

Prior to the nineteenth century, or as Altmann named it, “the pre-history of psycholinguistics”, it was controlled by “philosophical conjecture”. He means that era was lacking “systematic and ongoing questioning of the relationship between mind and language, or indeed, brain and language—there was no community of researchers asking the questions.” While experimental investigations (such as measuring reaction times, monitoring eye movements, and recording babies’ babbles) are the norm of modern-day psycholinguistics, this does not mean that there were not any experiments at all before the nineteenth century, but “there were isolated cases, generally of a kind that would not be tolerated in the modern age” [1] (p. 258).

Many researchers divided the developing history of psycholinguistics into two major eras: historical, and modern. The first era occurred around the beginning of the nineteenth century, whereas the second one took place during the end of the twentieth century [6,7]. The focus on the psychology of language changed, by the final decades of the nineteenth century, from language breakdown into its normal use. The importance of mental states and the connection between utterances and those internal states were emphasized by Wilhelm Wundt, who published *Die Sprache* in 1900 [1,6].

Wilhelm Wundt, known as the “father of experimental psychology”, considered language as the outcome of psychological processes; hence, important insights into the nature of mind could be revealed through studying the language. Wundt’s studies contributed to theories of both psycholinguistics and linguistics and highly influenced other researchers’ work, such as Hermann Paul (1846–1921), who based his work on Wundt’s thought [8].

Thereafter, the influence of *behaviourism* appeared in the early twentieth century. *Behaviourists* were against the Wundtian approach of consciousness and introspection and claimed that behaviour and behavioural observation should be the main interest of psychology. From this point of view, J.R. Kantor opposed the idea that language use implicated distinct mental states. That century witnessed a great shift in linguistics when Ferdinand de Saussure introduced structure into the study of language; for more, see [1] (p. 260). Then, in the 1930s, the Bloomfieldian school of linguistics emerged. Leonard Bloomfield transformed the linguistics’ perspective from the historical and comparative study of languages to the language description of grammar, and he was looked at as one of the pioneers of American structuralism [9]. The investigation of language structures was reduced by Bloomfield [10] “to a laborious set of taxonomic procedures, starting with the smallest element of language—the phoneme” [9] (p. 260). Hence, he firmly associated linguistics with the *behaviourism* approach, which asserts that language should be represented by visible and measurable behaviour as a set of stimuli and responses rather than mental states. In 1957, B.F. Skinner published “Verbal Behaviour”, which was regarded the end of the *behaviourism* [11]. Skinner tried to explain verbal learning and verbal behaviour in the light of conditioning theory using *behaviourist* principles [1].

The Chomskyan influence on psycholinguistics appeared in the mid of the twentieth century and his universal grammar viewed as a direct challenge to the *behaviourist* theories. The American linguist Noam Chomsky reviewed Skinner’s “Verbal Behaviour” in 1959 and debated that conditioned stimulus–response associations could not justify “the infinite productivity or systematicity of language.” Hence, Chomsky revolutionized linguistics and mental representation was introduced again into theories of language, which paved the way for the “cognitive revolution”. Through his work, Chomsky obviously showed that “language was founded on precisely mental representation”, in contrast with Skinner, who avoided these representations. Further, Chomsky clarified the way in which language is learned by children and distinguished between “competence”, i.e., the knowledge about a language, and “performance”, i.e., the use of that language [1,9].

Ayudhya and Kess [7,9] stated that Maclay in 1973 [12] classified the progression of modern psycholinguistics into four major periods: (1) formative; (2) linguistic; (3) cognitive; and (4) cognitive science or psycholinguistic theory or psycholinguistic reality. These are introduced briefly below.

### 1.2. Formative Period

In 1951, linguistics and psychology formally met for the first time at a seminar of Social Science Research at Cornell University. As a result of this meeting, a committee on linguistics and psychology was formed. The key issues investigated during this time were: how people comprehend and produce language, how and under what circumstances they lose language, how a particular language affects cognition, and the connection between the first and foreign language learning. During in the formative period, the dominant paradigm in linguistics was structuralism and the trend in psychology was *behaviourism*. Hence, both disciplines adopted a *behaviourism* approach [7,9,12].

### 1.3. Linguistic Period

Psycholinguistics was controlled during this period by generative grammar for the sake of studying language comprehension. There was a belief that understanding speakers’ competence gives a picture about the nature of speakers’ actual performance. According to the transformational generative grammar, the sentence has a significant role in explaining the grammar’s data and dimensions. Thus, they study the comprehension and use of sentences [7,9,12].

The American psychologist George A. Miller was one of the founders of cognitive psychology and cognitive neuroscience. His research contributed to psycholinguistics and the study of human communication and was considered a bridge between linguistic theory and psychological experimentation. In 1951, he wrote his first book entitled *Language and Communication*, which was regarded as a fundamental work in psycholinguistics. This period gradually encouraged a more interdisciplinary in psycholinguistics than in the formative period [13].

### 1.4. Cognitive Period

The dependence of language upon human cognition is the main premise behind the cognitive approach. Language is considered “one of several fundamental cognitive process outcomes” [9] (p. 144). Psycholinguistics was a branch of cognitive psychology and was entirely independent of linguistics. Thus, psycholinguistic research within this period examined language from the perspective of cognitive psychology. The best early examples of this approach are Bever [14] and Slobin [15]. Bever’s goal was improving the uses of human language through the application of linguistics, psycholinguistics, and cognitive science [14]. Slobin’s work also highlighted the significance of cross-linguistic comparison on the investigations of language acquisition and psycholinguistics in general [15]. During the 1980s, psycholinguistics involved studying first and other language learning, children’s language acquisition, and linguistic disabilities. Ayudhya mentioned three key questions of psycholinguistics research during this time [9]:(a)The study of the link between psychology and linguistics in mental representations and language processing;(b)The study of the language-processing processes that mental representation of speakers transformed from a process into another;(c)The study of overall language processing in which each level of language processing interacts with the other levels.

### 1.5. Psycholinguistics Theory Period

In the 1980s, there was a shift in psycholinguistics progression. None of psychology or linguistics dominated psycholinguistics. Moreover, because of the influence of cognitive science in the previous period, it was demonstrated that psycholinguistics can be studied as the scientific comprehension of the way in which the human mind is involved in how language works. Therefore, psychological reality can be investigated scientifically in psycholinguistic theory. In this period, cognitive psychology and linguistics merged, and psycholinguistic research contributed to a realistic science of the human mind [9].

Psycholinguistics has developed into an interdisciplinary field. That is, other disciplines such as biology, neuroscience, psychology, cognitive science, computer sciences, and language teaching and learning are applied to investigate language processing [16].

### 1.6. The Scope of Psycholinguistics

The section above leads us to the inevitable question: what is psycholinguistics? There are several definitions of this science, which are similar in certain aspects and different in others. These differences could arise from researchers’ different views and backgrounds and from their arrangement of its topics [4].

Psycholinguistics is an interdisciplinary field of study that can be defined as “the study of the mental representations and processes involved in language use, including the production, comprehension and storage of spoken and written language” [17] (p. 4). Warren proposed that psycholinguistics consists of language processes, production, and comprehension. Language processes are either central or peripheral. Language production includes intention, planning, lexicalization, and articulation. Language comprehension consists of perception, word recognition, parsing, and interpretation. He added that these processes undergo different linguistic levels: phonological, phonetic, morphological, syntactic, sematic, and then discourse analysis [17] (p. 5).

More interestingly, the American Psychological Association (APA) dictionary of psychology considers psycholinguistics as a branch of psychology and defines it as [18]:

A branch of psychology that employs formal linguistic models to investigate language use and the cognitive processes that accompany it. Developmental psycholinguistics is the formal term for the branch that investigates language acquisition in children. In particular, various models of generative grammar have been used to explain and predict language acquisition in children and the production and comprehension of speech by adults. To this extent, psycholinguistics is a specific discipline, distinguishable from the more general area of psychology of language, which encompasses many other fields and approaches.

Other researchers such as Issa and Awadh considered psycholinguistics as interdisciplinary branch of linguistics that concerns “the cerebral foundations of language usage” [19] (p. 20), which is obviously linked to areas of linguistic study (i.e., phonetics, phonology, morphology, syntax, semantics, and discourse analysis) [17], while others such as Fernandez and Cairns took a broader view and pertained psycholinguistics to psychology and linguistics as a sub-discipline of them both, yet emphasizing that it is also associated with developmental psychology, cognitive psychology, speech science, and neurolinguistics [20].

Psycholinguistics aims at understanding how individuals acquire language, how they use language to speak and understand each other, and how language is represented and processed in the brain [20]. Hence, we can say the core goal of this interdisciplinary field of study is developing a coherent theory about how humans comprehend and produce language [21,22].

Since the scope of this discipline is to investigate how language is used and learned [23], it is, as stated in the APA Encyclopaedia of Psychology [24], related to: the traditional academic disciplines of linguistics, psychology, education, the cross-disciplinary areas of speech science, cognitive science, artificial intelligence, neurolinguistics, language learning, teaching, and rehabilitation.

To sum up, psycholinguists study the psychological processes involved in the language usage, including language understanding, language production, and first and second language acquisition [25].

### 1.7. Scientific Contributions for Psycholinguistics

In this section, we attempt to shed light on scientific contributions made to psycholinguistics, i.e., journals and associations dedicated to the field. Table 1 shows a breakdown of the data. We should point out that we only included journals, associations, and research centres with the words “psycholinguistics” or “psycholinguistic” in their titles.

### 1.8. Purpose of the Present Study

Psychology has been credited with advancing the study of human thought and behaviour, including language, in past reviews of psycholinguistics [26]. In the late 19th century and early 20th century, *behaviourism* dominated the explanation and research evidence concerning the psycholinguistic processes associated with human language. In the second half of the 20th century, however, this changed with the rise of Chomsky’s theory of language [27]. With the advancement of technology during the early 21st century, the study of psycholinguistics shifted to a computational model of language and speech processing using a connectionist approach [28]. There was also an increase in cross-linguistic studies aimed at identifying universal aspects of children’s language development, usage, and breakdown across languages [29].

As the study of psycholinguistics progresses, many researchers have criticized the focus on monologues in empirical evidence rather than the sociocultural implications of dialogue-based evidence [30]. There has been an expansion of psycholinguistics to examine more specific aspects of human language, such as the production of language [31] and integrating the study of language with the cognitive sciences [32]. Recent reviews of psycholinguistics have focused on the use of offline measures of language comprehension [33], psycholinguistics and teaching and learning settings to enhance education [34], and the syntactic and cognitive aspects of language acquisition and learning [35]. Recent bibliometric analyses examined data from 1900 to 2021 concerning child language, which is the subject of psycholinguistic research. However, this review limited its data collection to articles and only used the *WOS* [36].

In this study, we examine the development of the field of psycholinguistics and the evidence regarding how (children) acquire, learn, process, comprehend, and produce language. The difference between this study and previous ones is that it is more comprehensive in that it includes data from 1946 to 2022. As part of the triangulation process, three databases were used (*Scopus*, *WOS*, and *Lens*) to ensure the data are not biased towards particular journals within the field. In addition, it combines bibliometric and scientometric indicators to analyse state-of-the-art psycholinguistic scholarship. As a result, this study seeks to apply the science mapping approach [37] “to detect and visualize emerging trends and radical changes” [38] (p. 374) and patterns in literature [39] pertaining to psycholinguistics.

It should be noted that our research in this study is restricted to articles with the term “psycholinguistic*” in their title, abstract, author keywords, and topic. Examining the use of “psycholinguistics” or “psycholinguistic” concepts is the raison d’être of this exclusion. This means that, while we are aware that numerous other topics and themes (e.g., first language acquisition, second language acquisition, language learning, child language, etc.) are within the scope of psycholinguistic research, we avoided making our resarch lengthy and instead focused on the use and development of the two concepts listed above. The following questions therefore guided our research: (1) What is the size of psycholinguistics’ knowledge production over the past seventy-six years as measured by year, region, institutions, journal, publisher, research area, author, and cited documents? (2) Who are the most influential and central authors in the field of psycholinguistics? (3) What are the most sought-after terms and keywords in psycholinguistics? (4) Which patterns are the most investigated and studied in psycholinguistics?

## 2. Methods

### 2.1. Research Methods

Scientometrics is the methodology for examining artifacts or objects; one examines not the process of science and scholarship but the outcome of these activities [40] (p. 491). Scientometrics studies “the quantitative aspects of the production, dissemination and use of scientific information with the aim of achieving a better understanding of the mechanisms of scientific research as a social activity” [41] (p. 6). Research of this type may aim to improve the quality of publications, but its purpose is not entirely clear. According to previous research, “the task of determining quality papers is especially difficult in BIS (bibliometrics, informetrics and scientometrics) due to the very heterogeneous origin of the researchers” [42] (p. 390). Nevertheless, these studies aim to “reveal characteristics of scientometric phenomena and processes in scientific research for more efficient management of science” [43] (p. 1).

A scientometric indicator guides studies of this nature. A number of factors are taken into account (e.g., publications, citations and references, potential, etc.) as well as type indicators [43]. The most common method used in such studies is “mapping knowledge domains”, which refers to creating “an image that shows the development process and the structural relationship of scientific knowledge” and using maps that are “useful tools for tracking the frontiers of science and technology, facilitating knowledge management, and assisting scientific and technological decision-making” [44] (p. 6201). It is currently becoming more common for research of this type to include all fields of study without limiting itself to medical, health, and pure sciences [45]. This study examines psycholinguistics as a sub-field of linguistics that integrates psychology and linguistics.

### 2.2. Measures

It is widely acknowledged that studies in bibliometrics and scientometrics are valuable tools for guiding the assessment of knowledge produced in the field or concept under consideration (e.g., psycholinguistics). Most knowledge databases (e.g., *Scopus*, *WOS*, and *Lens*) contain bibliometric indicators [46,47,48,49]. Scientometric indicators are provided by a scientometric program. This study used CiteSpace 5.8.R3 [50] and VOSviewer 1.6.18 [51]. Table 2 lists the bibliometric and scientometric indicators used in this study.

### 2.3. Data-Collection and Sample

The data were retrieved using *Scopus*, *WOS*, and *Lens*. These databases were included for a variety of reasons. *WOS* and *Scopus* initially cover only publications that match their criteria [46,47,48]. In addition, *Lens* includes more data than both *Scopus* and *WOS* [49]. In other words, while both *Scopus* and *WOS* only listed publications that are included in their indexed journals, *Lens* includes more data beyond these two databases. Although *Google Scholar* might have more data than *Lens*, it is not as systematic as *Lens*, and data retrieval for large sets of data from *Google Scholar* is smooth.

The search was conducted on Saturday, 11 June 2022. There were no language restrictions if titles, abstracts, and keywords were written in English. A limited number of publications were available in other languages, so the results were manually verified. Among the types of publications considered were articles, book chapters, book reviews, and conference proceedings (full papers), including early access publications of these types. The search strings and other specifications for the three databases are listed in Table 3.

We examined how “psycholinguistics” or “psycholinguistic” is used to describe the size and change of research in this field. Thus, the keywords we used did not include any terms specific to a particular age group, type of learner, language, or any other expanding concept since the above keywords returned a large number of publications.

### 2.4. Data Analysis

Analysing the data involved a number of steps. Data from *Scopus* were initially exported in three formats: Excel sheets for bibliometric analysis, RIS for CiteSpace, and CSV for VOSviewer. To work with CiteSpace, it was necessary to convert the RIS file into a *WOS* file. In addition, *WOS* data were extracted in two formats: text files converted to Excel sheets for bibliometric analysis and plain text files for CiteSpace and VOSviewer. Our last step was to obtain *Lens* data in two formats: CSV for bibliometric analysis and full-record CSV for VOSviewer.

Prior to CiteSpace analysis, duplicate documents were eliminated using CiteSpace and Mendeley. The bibliometric analysis was performed using Excel. Citation reports were generated using Excel and converted into figures.

For scientometric analysis, the software packages were set to the default settings. For each database, separate visualizations were developed, including network visualizations, overlay visualizations, and density visualizations. Each analysis was performed three times for *Scoups* and *WOS*: by author keyword, by source, and by author cited. We performed four analyses for *Lens*: cooccurrence analysis by author keywords, (co)citation analysis by author, (co)citation analysis by source, and (co)citation analysis by document. The following analyses were conducted in CiteSpace for *Scopus* and *WOS*: co-citations by document (references), co-citations by cited authors, and occurrence (keywords). Results included narrative summaries, cluster summaries, visual maps, and burst tables.

## 3. Results

### 3.1. Result Overview

Presented below are two sections of our results. Bibliometric indicators for psycholinguistics are presented in the first section. Data from *Scopus*, *WOS*, and *Lens* databases were used to calculate the indicators. These bibliometric indicators include publications by year, top 10 countries, universities, journals, publishers, subject areas, and authors. The second section provides scientometric indicators relating to the development of psycholinguistics. CiteSpace and VOSviewer were used to analyse the indicators. Citation and co-citation indicators are included as well as cooccurrence indicators.

### 3.2. Bibliometric Indicators for the Study of Psycholinguistics

#### 3.2.1. Production of Psycholinguistics Knowledge by Year

A total of 12,172 from *Scopus*, 4845 from the *WOS*, and 15,551 from *Lens*, psycholinguistics documents were retrieved for analysis. The data periods for the three databases were 1946–2022, 1985–2022, and 1946–2022. The documents from *Scopus* included 9952 articles, 600 review articles, 567 book chapters, 263 books, and 781 conference papers. The documents from the *WOS* included 4056 articles, 222 review articles, 186 book (chapters), 79 early access, and 671 proceedings papers. The documents from *Lens* included 7755 articles, 3796 unknown, 1242 book chapters, 1084 books, 499 dissertations, and 744 conference proceedings (articles) and preprints. Most of these documents were in English with the inclusion of other languages such as Russian, Spanish, French, Portuguese, German, etc. Since the analysis is based on the title, keywords, abstract and references, they all include this information in English. To avoid bias, this inclusion was neutral regarding published English-language data.

Figure 1A–C shows the length of production by year for the three databases. As can be seen, psycholinguistics has experienced a significant increase in knowledge production, reaching its peak in 2019 with 757 publications in *Scopus*, 419 publications in the *WOS*, and 929 publications in *Lens*. The *Scopus* publication range is 1–757, the *WOS* is 6–419, and *Lens* is 1–929. The total documents analysed were 32,586 of which 25,621 were published between 2000 and 2022. All databases have the lowest number of publications in the previous year. Thus, the production of knowledge related to psycholinguistics has significantly increased in the past two decades.

#### 3.2.2. Production of Psycholinguistics Research by Country and University

Figure 2A–C shows the top 10 producing countries for knowledge related to psycholinguistics. As can be seen, the top 10 countries all belong to the Western world except for Russia, which appears in three databases, along with China and Brazil in *Lens*. Even though these countries dominate the production of psycholinguistics research, contributions extend beyond this list to the countries after the top 10.

Figure 3A–C presents the top 10 universities and/or research centres producing knowledge in psycholinguistics. Most of these universities or research institutions are in Europe, with British universities having significant representation in all three databases. The Max Planck Institute of Psycholinguistics and Max Planck Society appear to have a high production level of knowledge related to psycholinguistics.

#### 3.2.3. Production of Psycholinguistics Research by Journal and Publisher

Figure 4A–D demonstrates the top 10 journals publishing research in psycholinguistics. Several journals include the word “psycholinguistics” or “psycholinguistic” in their titles. There are also other terms close to psycholinguistics, including the word “psychology”. An extended list of journals based on publishers is shown in Figure 4D. A few journals are related to medical and health sciences (e.g., *Aphasiology*), and cognitive science (e.g., *Human Cognitive Processing*).

Figure 5A,B shows the list of top 10 publishers for knowledge in psycholinguistics. As *Scopus* does not include publisher information, these lists are limited to *WOS* and *Lens* databases. In terms of publishing research related to psycholinguistics, Springer Nature and Elsevier achieve the top two rankings in both databases.

#### 3.2.4. Production of Psycholinguistics by Research Area, Keywords, and Cooccurrence

In psycholinguistics, psychology and language are integrated, but they also integrate with other fields (Figure 6A–C). Psycholinguistics is dominated by publications in psychology, arts and humanities, social sciences, and neuroscience, as shown in Figure 6A. According to Figure 6B, linguistics, psychology, neurosciences, and neurology constitute the top four research areas in psycholinguistics. Several of these findings are confirmed in Figure 6C, where psychology, linguistics, psycholinguistics, and computer science are introduced as the four top fields of study in psycholinguistics. *Lens* shows more specific fields associated with this field of study (e.g., language acquisition, language use, perception, and comprehension).

#### 3.2.5. Production of Psycholinguistics by Authors

It should never be confined to only top authors in the field to contribute to psycholinguistics because any paper published in the field is a contribution. Our goal was to display authors that produced more psycholinguistic knowledge as shown in (Figure 7A–C). Although the ranking differs according to the database, most of the authors are the same across the three databases. Gibbs [56], who is ranked first in *Scopus* and *WOS*, is ranked ninth in *Lens*. It could be due to *Lens’* comprehensiveness to include more data compared to *Scopus’* and *WOS’* limitation to indexed journals and documents.

### 3.3. Scientometric Indicators for the Study of Psycholinguistics

#### Overview of Psycholinguistics Studies from *Scopus*, *Web of Science*, and *Lens*

In this section, scientometric analysis is presented for the data retrieved from *Scopus*, *WOS*, and *Lens* databases. Psycholinguistics is discussed by highlighting certain concepts, authors, references, and emerging trends.

CiteSpace was used to show the top keywords with the strongest citation bursts for *Scopus* and *WOS* data (Figure 8A,B). All research is represented by the green line. Red lines indicate the start and end of bursts. The word with the strongest citation burst in *Scopus* is (support = 229.89) between 1975 and 1995, and (children = 14.72) between 1993 and 2001 for the *WOS*. There is a variation in the citation burst depending on the database used. *WOS* contains retrieval, aphasia, deficit, etc., but *Scopus* only has attention, young adult, procedure, etc.

Visualisations of clusters and authors further illustrate these concepts (Figure 9A–D). Topics like psycholinguistic abilities and eye movement are among the most explored topics in psycholinguistics, as shown in Figure 9A. Figure 9B illustrates more specific concepts, including sentence processing, speech production, and language acquisition. Figure 9C,D shows the most-cited references and the topics searched while citing them. Bilingual language control, ERP correlates, etc., are among these topics (see Figure 9C). Other words included in the WOS database include cortical dynamics, language learning, and naming sentences. (See Figure 9D). It should be noted that next to each cluster, the authors and relevant works are highlighted, and the more intense the text is, the more popular the cluster will be.

It is also important to consider the cooccurrence of keywords used. We generated three visual network maps for the most frequently used psycholinguistic keywords in the three databases using VOSviewer (Figure 10A–C). Each colour represents a different direction in psycholinguistics. A green topic refers to imaging techniques, a blue one to bilingualism, an orange one to word recognition, and a red one refers to psycholinguistics (see Figure 10A). These colours change according to the database. In Figure 10B, yellow indicates discourse analysis, orange indicates assessment, and green indicates psycholinguistics. Language processing and comprehension keywords are highlighted in red in Figure 10C.

We generated three visual network maps for co-citation and citation by author using VOSviewer (Figure 11A–C). For each colour, there is a network of co-citations or citations for the authors. The larger the circle, the more co-cited or cited the author is. Similar authors appear in all three databases whether they are co-cited or cited. These include Baayen [57], Chomsky [58], Freiderici [59], Bonin [60], etc.

We generated three network maps of co-citations and citations by source using VOSviewer (Figure 12A–C). Colours represent networks of co-citations or citations. Circle size indicates how many times the source has been co-cited or cited. According to Figure 12A, *Journal of Memory and Language*, *Neuroimage*, and *Journal of Cognitive Neuroscience* are the most co-cited sources. Figure 12B shows comparable results using the *WOS* database with more significant journals (e.g., *Brain and Language*). The citation network for journals is shown in Figure 12C. These include *Frontiers in Psychology*, *Journal of Psycholinguistic Research*, *Psycholinguistics*, etc.

The top 10 cited works were extracted from *Scopus*, *WOS*, and *Lens* bibliometric data. Our next step was to merge them and to remove duplicates from them (Table 4). Citations are reported next to each document and whether they were reported in all three databases.

### 3.4. Impact of Research on Psycholinguistics by Clusters, Citation Counts, Citation Bursts, Centrality, and Sigma

#### 3.4.1. Clusters

The network is divided into 12 co-citation clusters (Table 5 provides details). The largest 4 clusters are summarized as follows. The largest cluster (#0) has 204 members and a silhouette value of 0.914. It is labelled as affective norm by LLR, individual difference by LSI, and familiarity account (3.15) by MI. The most relevant citer to the cluster is Veldre [85]: “Semantic preview benefit in English: individual differences in the extraction and use of parafoveal semantic information”.

The network is divided into 20 co-citation clusters (see Table 5 for further details). The largest eight clusters are summarized as follows. The largest cluster (#0) has 131 members and a silhouette value of 0.896. It is labelled as formulaic language by both LLR and LSI and as persistence (1.76) by MI. The most relevant citer to the cluster is Smith [86]: “The effect of word predictability on reading time is logarithmic”.

#### 3.4.2. Citation Counts

In *Scopus*, the third is Baayen [57] in Cluster #3, with citation counts of 54. The fourth is Kutas [87] in Cluster #5, with citation counts of 45. In the *WOS*, the third is Baayen [57] in Cluster #0, with citation counts of 41. The fourth is Kuznetsova [88] in Cluster #2, with citation counts of 41. Table 6 lists the remaining top authors based on citation counts.

#### 3.4.3. Bursts

In *Scopus*, the third is Baayen [57] in Cluster #3, with bursts of 24.53. The fourth is Kutas [87] in Cluster #5, with bursts of 20.76. In the *WOS*, the third is Baayen [57] in Cluster #0, with bursts of 20.62. The fourth is Kuznetsova [88] in Cluster #2, with bursts of 20.30. See Table 7 and Figure 13A,B for the remaining top 10 detected bursts in psycholinguistics.

#### 3.4.4. Centrality

In *Scopus*, the top ranked item by centrality is Baayen [57] in Cluster #3, with centrality of 58. The second one is Pickering [94] in Cluster #5, with centrality of 46. In the *WOS*, the third is Warriner [97] in Cluster #1, with centrality of 39. The fourth is Alonso [103] in Cluster #1, with centrality of 38. The remaining top central authors in psycholinguistics are listed in Table 8.

#### 3.4.5. Sigma

In *Scopus*, the top ranked item by sigma is Baayen [57] in Cluster #3, with sigma of 0.00. The second one is Pickering [94] in Cluster #5, with sigma of 0.00. In the *WOS*, the third is Warriner [97] in Cluster #1, with sigma of 0.00. The fourth is Alonso [103] in Cluster #1, with sigma of 0.00. See Table 9 for the remaining authors receiving high attention by researchers in psycholinguistics.

## 4. Discussion

The current study sought to assess the evolution of knowledge in psycholinguistics, an interdisciplinary field that studies how language is used and learned [23]. This objective was achieved by tracing the history of psycholinguistics and presenting bibliometric and scientometric indicators for 32,568 documents published over the past 76 years. Two sections were devoted to presenting the results. The first section included bibliometric indicators such as publications by year, the top ten nations, universities, journals, publishers, subject areas, and authors. Indicators for scientometrics, including citation, co-citation, and cooccurrence, were presented in the second section.

The study’s key findings include the following seven points: (1) The production of knowledge in clinical linguistics increased in the last two decades, where of 32,586 analysed documents, 25,621 were published between 2000 and 2022. (2) While the U.S. leads in all the three databases, followed by the U.K. and Germany, the situation slightly changes if we consider the top 10 universities rankings. In this case, (3) U.S. universities are superseded by European ones. (4) The top journals publishing in the field are the *Journal of Psycholinguistic Research*, *Psycholinguistics*, and *Behaviour Research Methods*, (5) while Elsevier and Springer Nature are the major publishers. (6) There are a plethora of subject areas related to psycholinguistics. Among these we find psychology, arts and humanities, social sciences, neuroscience, linguistics, and computer science. The last finding (7) shows that Gibbs [56], Pulvermuller [106], and Rickheit [107] are some of the major authors contributing to the field of psycholinguistics.

When linked to the scientometric findings, the above findings have at least five implications. First, by identifying the most-searched terms, researchers can be led to the most contentious topics and themes in psycholinguistics research. In this study, they incorporated attention [108], child [109], physiology [110], young adult [111], and procedure [112]. They also included aphasia [113], deficit [114], prediction [115], constraint [116], and French [117].

The second implication refers to identifying patterns and tendencies regarding the structure and dynamics of psycholinguistics-related knowledge. This could be accomplished through co-citation calculation. Importantly, since we analysed more than 32,000 documents containing only the term “psycholinguistic*”, clustering was used to organise this massive set of data. In addition, our analysis uncovered the largest co-citation clusters. Among the clusters were individual difference [118], sentence processing [119], formulaic language [120], and Spanish word [121]. Overall, there were 12 clusters summarised as follows:Examining individual difference in affective norm and familiarity account;Examining the refractory effect in the role of Broca’s area in sentence processing;Using eye movement to study bilingual language control and familiarity account;Exploring familiarity account through relative clauses;The study of formulaic language and language persistence;Examining affective norm and sub-lexical effect in Spanish words;Examining lexical persistence in multiplex lexical networks;The study of persistence through cortical dynamics;The study of context effect in language learning and language processing;The study of neurophysiological correlates in semantic context integration;Examining persistence as an acquisition norm through naming latencies; andFollowing a cross-linguistic perspective to study aphasic speakers.

Finding the authors who are frequently cited and have a significant impact on the direction of psycholinguistics research is the third implication. It is obvious that every contribution to the field of psycholinguistics adds to it, but the authors who receive the most citations are likely to have more in-depth knowledge of the subject than less-cited authors. Regarding the most recent works by the authors who receive the most citations, some of them, including Rickheit [107], Herrmann [122], and Brysbaert [91], address general topics such as the history of psycholinguistics [107], language use [122], reading [123], and individual differences [124,125]. Other authors, such as Levelt [3], address more specific topics, including visual word recognition [126], semantic frames [127], bilingualism [128], and grammar learning [129].

The fourth implication corresponds to the most frequently cited publications, which are important sources for psycholinguistics research. The majority of the articles were reviews of scientific literature that covered subjects such as latent semantic analysis [62], dual route cascaded model [71], reading, and word association [84]. A closer look at the identified 24 top-cited documents can lead to two patterns. First, the top-cited documents between 1967 and 2000 included speech perception [77], concreteness [68], psycholinguistic database (e.g., *MRC*, *WordNet*) [73,75,81], mental representation [80], retrieval in sentence production [63], word association and mutual information [84], second language acquisition [79], impaired vs. normal word reading [83], cross cultural psychology [69], auditory cortex and silent reading [64], knowledge representation and semantic analysis [62], phonological loop in language learning [82], and learning using the cognitive approach [61]. The second patterns included topics from the top-cited documents published after 2000, and these included topics such as using computational models for visual word recognition and reading [71], functional anatomy of language [70], reading acquisition [78], psycholinguistic models and neurobiological aspects of language [76], sentence production using abstract syntax [66], bilingual language production [67], sentence comprehension [72], word frequency measurement [74], and research in second language acquisition [65].

The final implication has to do with finding authors whose works in psycholinguistics might attract the attention of other authors and speedily become more frequently cited. Sigma metrics were used to calculate this component. Among the most-cited items stand out linear mixed models [57,89], emotional language [97,104], and language production and comprehension [94]. Even though our analysis highlighted assorted topics, they all related to psycholinguistics to some extent. For instance, linear mixed models are necessary to analyse nested data, whereas emotional language and production and comprehension are more related to the use of language in daily situations.

### 4.1. Practical Implications

Researchers should interpret the findings of scientometric studies carefully [130] no matter how popular this research method has become [131,132]. In this study, we retrieved data from multiple sources and avoided limiting ourselves to one database unless it was well-justified (i.e., *Scopus*, *WOS*, and *Lens*). To include various scientometric indicators in the analysis, different tools should be used (for example, both CiteSpace and VOSviewer were used in this study).

### 4.2. Theoretical Implications

There at least two practical implications in this study. The first reason for the elevated level of research in psycholinguistics is due to its integration with the education of languages, which makes it one of the first choices for researchers to explore and examine. There is, however, a need for more experimental and neuroscience-based research that will result in more credible findings of use to society and the advancement of educational psychology and the teaching of languages. Secondly, higher education institutions worldwide need to shift the focus of linguistics studies from their theoretical to scientific aspects. Psycholinguistics courses could be introduced, and even undergraduate and graduate degrees could be granted in the field. Studying theoretical linguistics should be a means to the study of language science, a degree in language sciences that has applied personal, societal, and economic value to the individual and the state.

### 4.3. Limitations

There are at least two limitations to this study. The first limitation pertains to the search string we used to retrieve the data. We searched for several types of documents with “psycholinguistics” or “psycholinguistic” in the title, abstract, author keywords, or topic. While this is justified by our intention to specifically track the usage of the term “psycholinguistics” and to avoid a lengthy paper, the inclusion of other specific topics may have altered these results. For example, we are fully aware that if we conducted a search for “language acquisition”, “sentence production”, and a variety of other terms, there will be large data sets for each of these search strings. The cluster analysis also has a limitation. Although we were able to cluster more than 32,000 documents into 12 psycholinguistics-related patterns, it was beyond the scope of our research to examine and present these clusters in detail.

### 4.4. Conclusions

This study examined the evolution and application of the term “psycholinguistics” over the past seventy-six years. Therefore, we conducted a scientometric study using eight bibliometric and eight scientometric indicators to analyse 32,586 *Scopus*, *WOS*, and *Lens* database documents. CiteSpace and VOSviewer were utilised to visualise and tabulate the analysed data, respectively. The principal findings comprised the presentation of visualisations and tabulations to produce knowledge in psycholinguistics by year, region, institution, journal, publisher, subject area, author, and most-cited documents. For example, of the 32,586 analysed documents, 25,621 were published between 2000 and 2022, indicating a significant increase in the production of knowledge in psycholinguistics over the past two decades. Importantly, we identified the most sought-after keywords, central authors, and those who may in the future receive more citations in the field of psycholinguistics. The grouping of 32,586 documents into 12 clusters demonstrating the major psycholinguistics research patterns is a crucial finding.

## Figures and Tables

**Figure 1 children-09-01471-f001:**
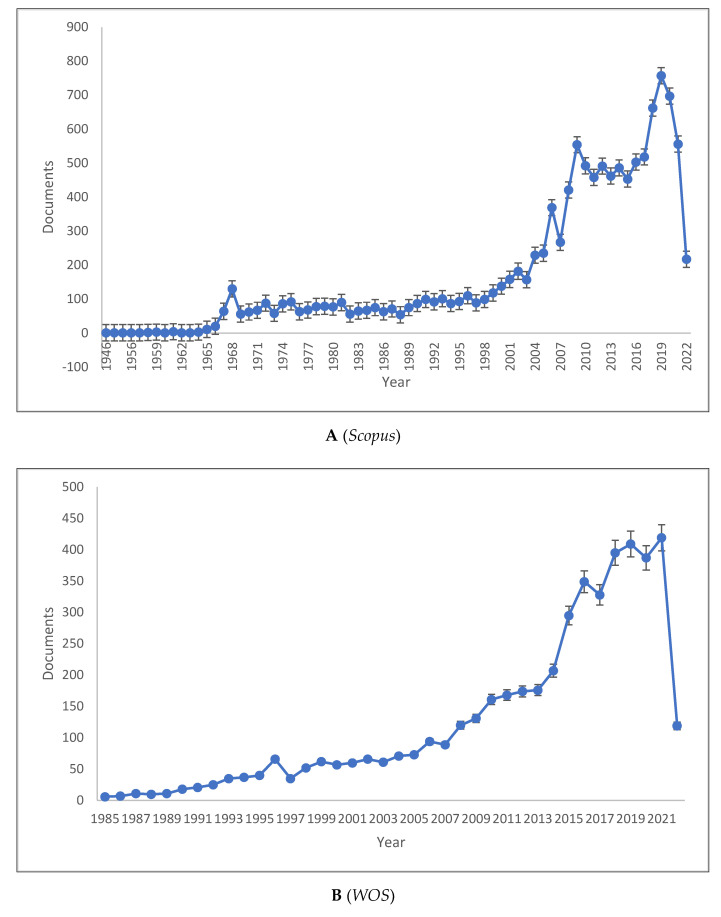
Psycholinguistics Knowledge Production Size by Year.

**Figure 2 children-09-01471-f002:**
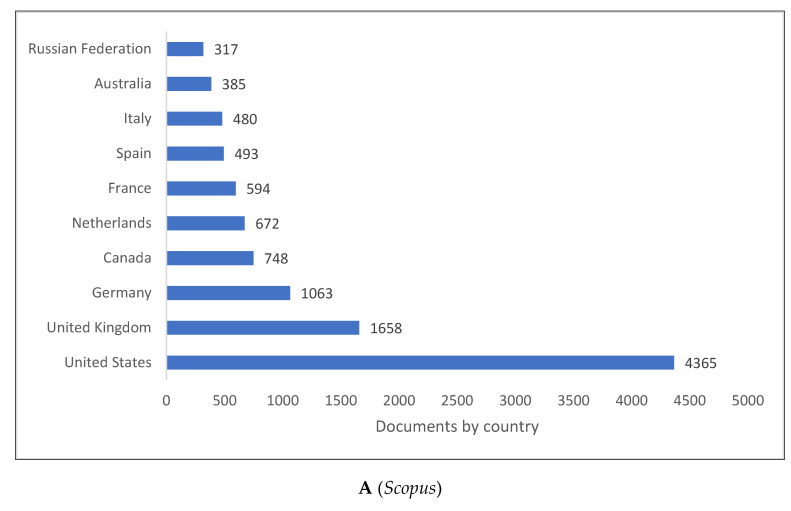
Psycholinguistics Knowledge Production Size by Country.

**Figure 3 children-09-01471-f003:**
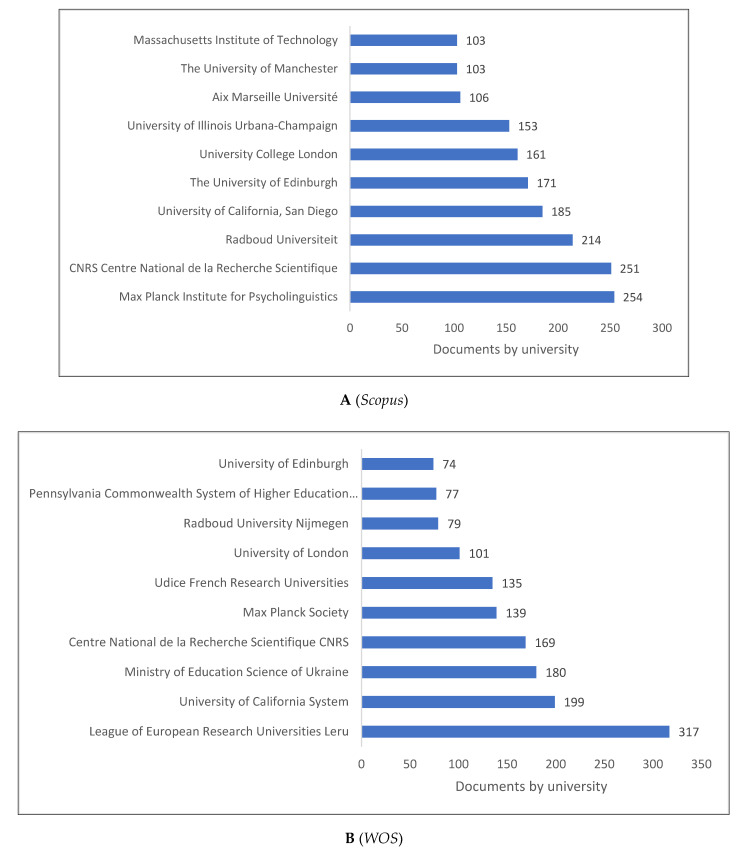
Psycholinguistics Knowledge Production Size by University/Research Centre.

**Figure 4 children-09-01471-f004:**
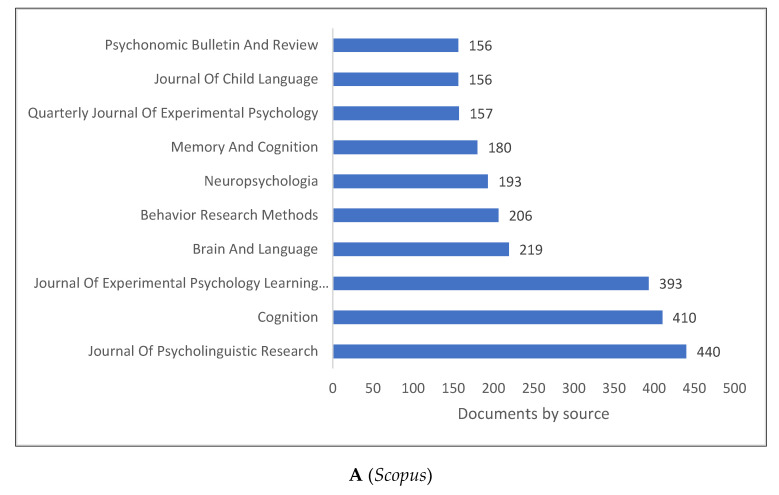
Psycholinguistics Knowledge Production Size by Journal.

**Figure 5 children-09-01471-f005:**
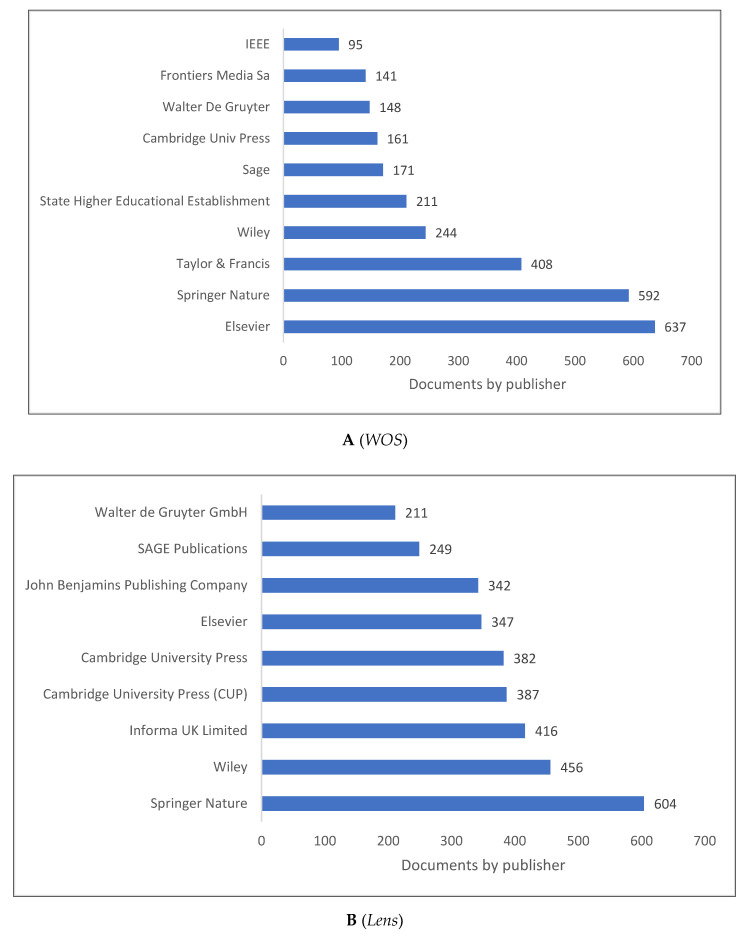
Psycholinguistics Knowledge Production Size by Publisher.

**Figure 6 children-09-01471-f006:**
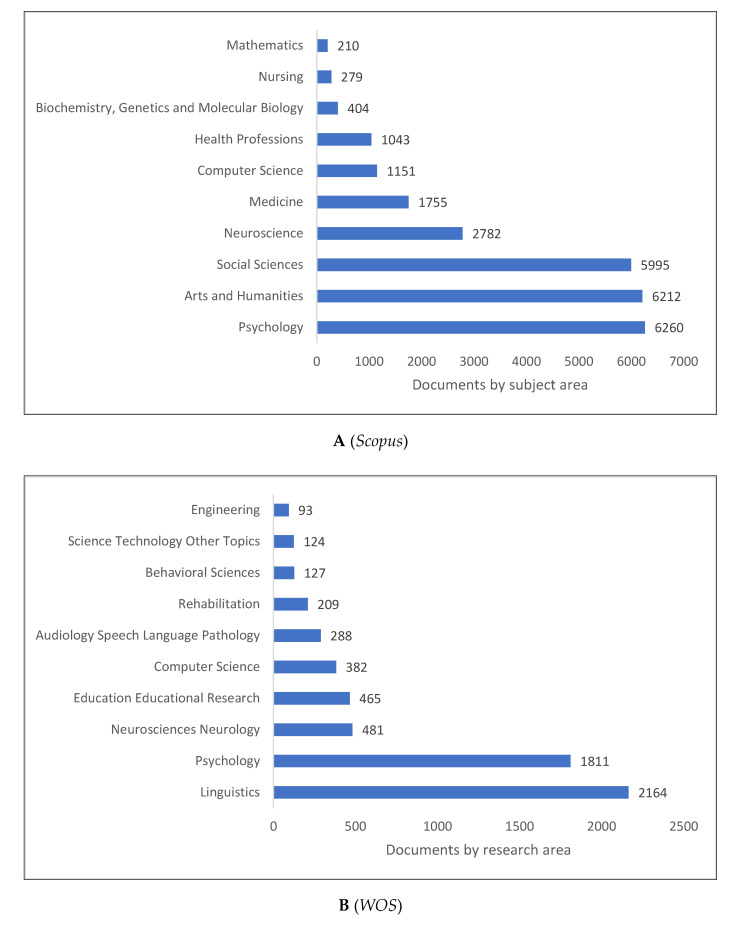
Psycholinguistics Knowledge Production Size by Research Area.

**Figure 7 children-09-01471-f007:**
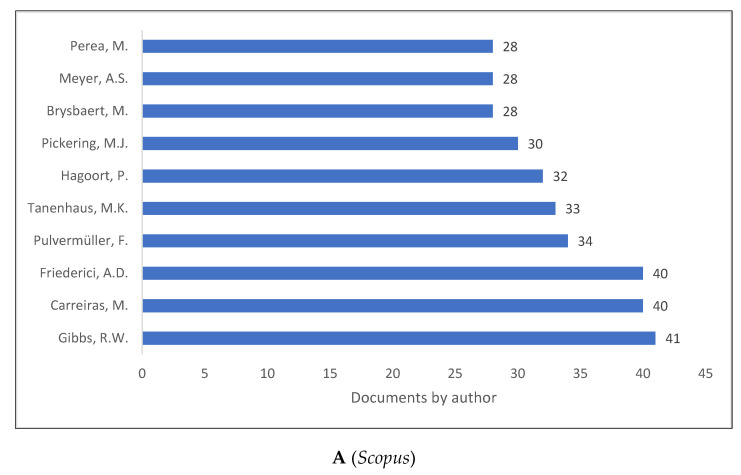
Psycholinguistics Knowledge Production Size by Research Area.

**Figure 8 children-09-01471-f008:**
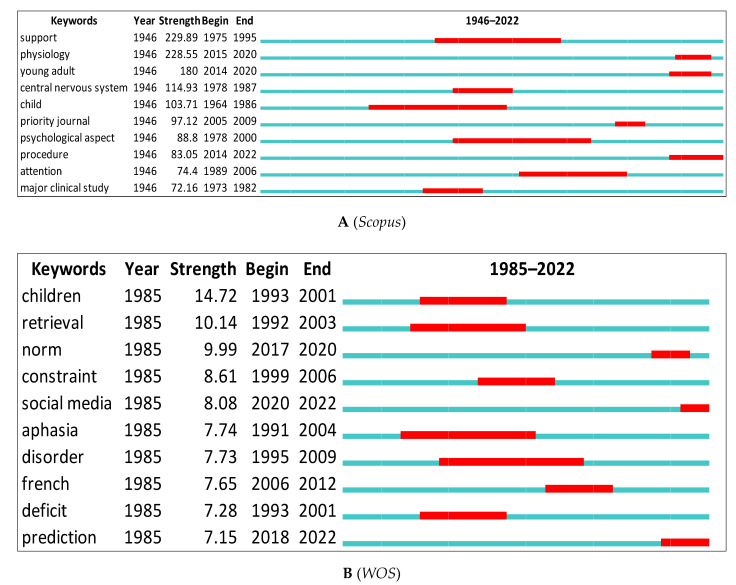
Top 10 Keywords with the Strongest Citation Bursts.

**Figure 9 children-09-01471-f009:**
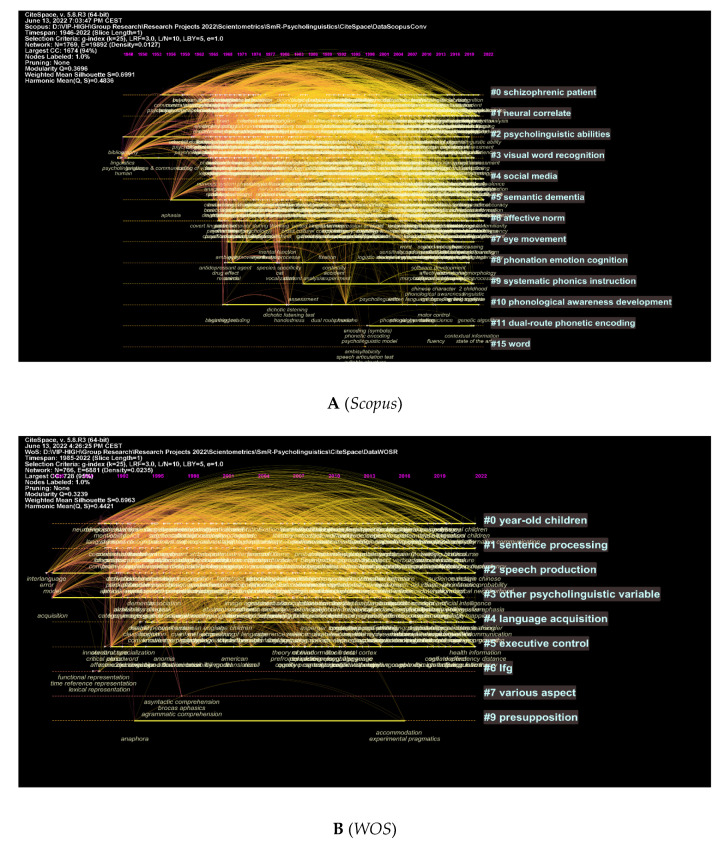
Top Keywords, Associated References, and Clusters.

**Figure 10 children-09-01471-f010:**
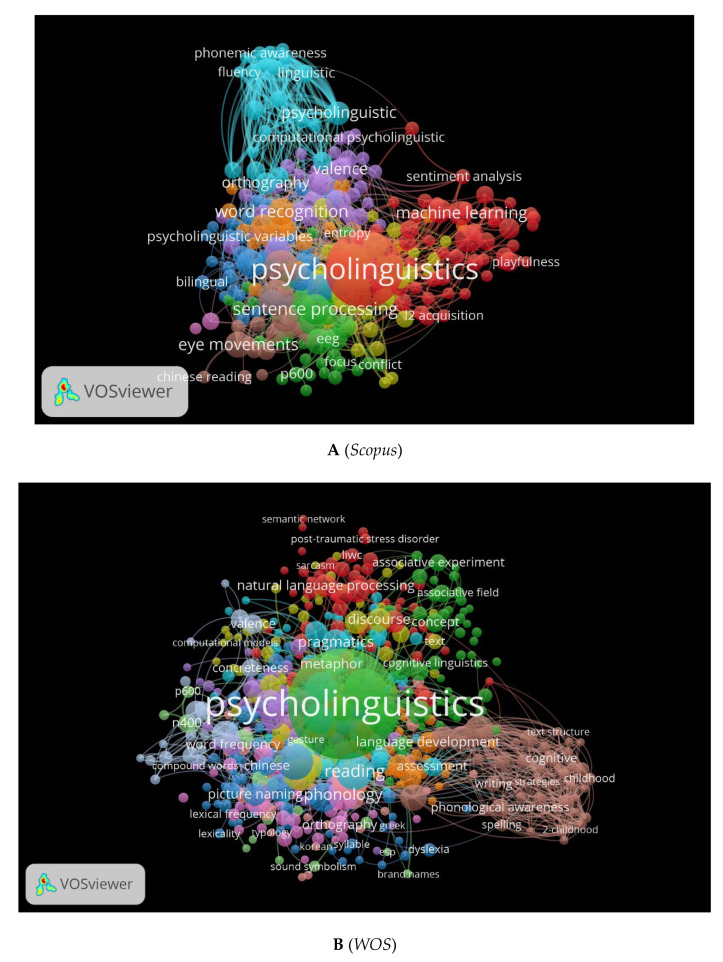
Cooccurrence by Author Keywords Network Visualization.

**Figure 11 children-09-01471-f011:**
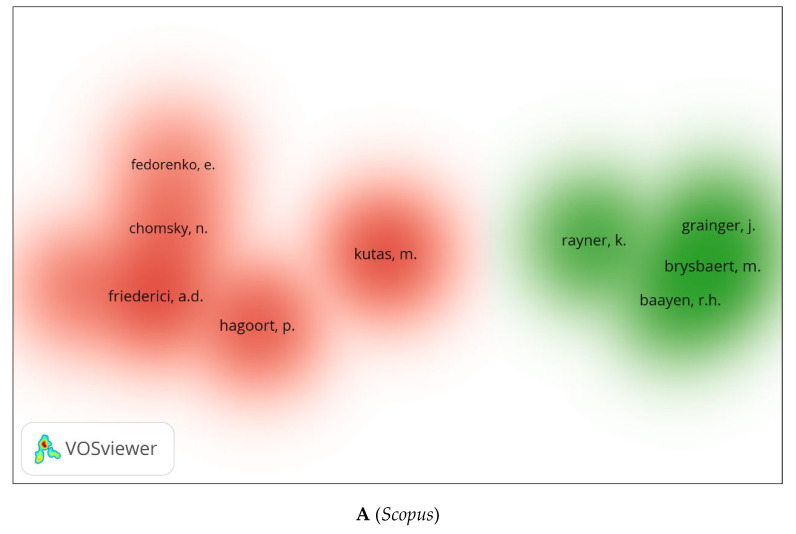
Co-citation by Cited Author Density Visualization.

**Figure 12 children-09-01471-f012:**
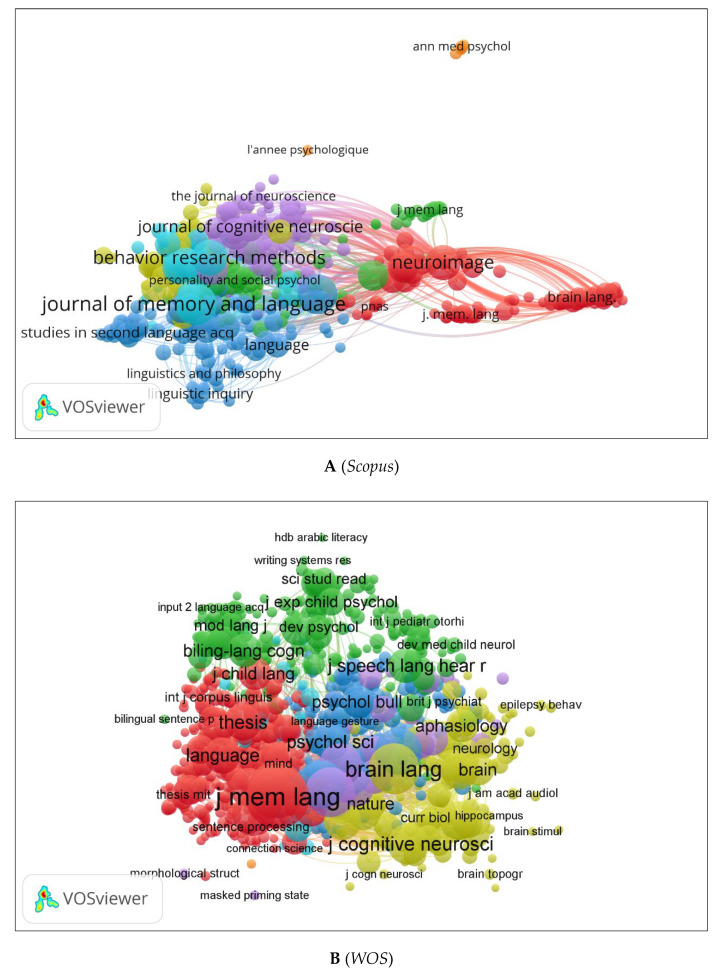
Co-citation by Source Network Visualization.

**Figure 13 children-09-01471-f013:**
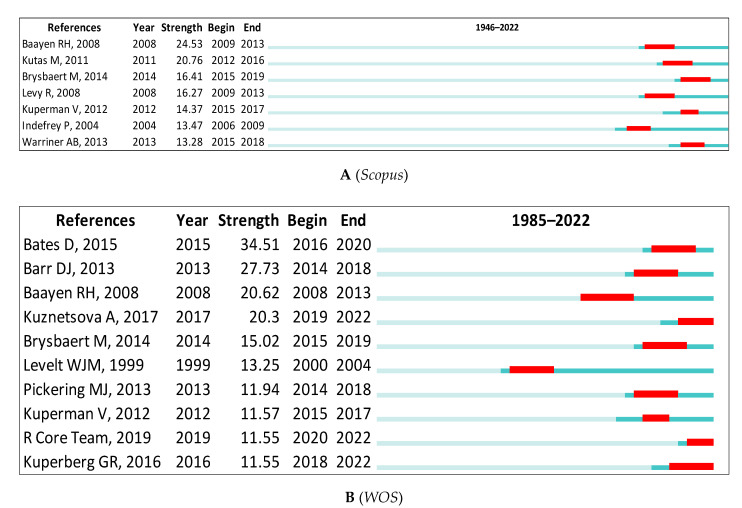
Top Seven References with the Strongest Citation Bursts [57,72,87,88,89,90,91,93,94,95,97,99,101,102].

**Table 1 children-09-01471-t001:** Psycholinguistics Journals, Associations and Research Centres.

No.	Journal	Started	Volumes Till Now	Scope	Website
1	*Applied Psycholinguistics*	1980	43	Psychological processes included in language, language development, language use, and language disorders in adults’ and children’s linguistics, psychology, reading, education, language learning, speech and hearing, and neurology.	https://www.cambridge.org/core/journals/applied (accessed on 15 April 2022)
2	*Journal of Psycholinguistic Research*	1971	51	Disciplines related to psycholinguistic research, the study of the communicative process as: the social and anthropological bases of communication; development of speech and language; semantics (problems in linguistic meaning); and biological foundations. In addition to the psychopathology of language and cognition, neuropsychology of language and cognition.	https://www.springer.com/journal/10936/ (accessed on 15 April 2022)
3	*Psycholinguistics*	2008	31	Production and perception of utterance and text, language consciousness, metalinguistic, linguistic, language and communicative competencies, formation, and development of verbal consciousness, the conscious and the unconscious in acquisition of languages, development of a language and communicative personality.	https://psycholing-journal.com/index.php/journal (accessed on 15 April 2022)
4	*East European Journal of Psycholinguistics*	2014	8	Bilingualism, clinical psycholinguistics, cognitive linguistics, cognitive psychology, discourse analysis, forensic linguistics, first and second/foreign language acquisition, neurolinguistics, psychology of language, and speech and translation studies.	https://eejpl.vnu.edu.ua/index.php/eejpl (accessed on 15 April 2022)
Associations/research centres					
1	International Society of Applied Psycholinguistics (ISAPL)	1982	Milan, Italy	Using psycholinguistic studies, research, and theoretical and methodological issues to solve practical problems.	https://uia.org/s/or/en/1100032583 (accessed on 15 April 2022)
2	African Psycholinguistics Association (APsA)	2019	South Africa	Researchers working on psycholinguistic topics are brought together.	http://apsa.africa/ (accessed on 15 April 2022)
3	Psycholinguistic Association of India (No clear information founded about this association)			Bringing together researchers who are investigating the interaction between language and psychological processing or plan to do so in the future.	http://www.worldcat.org/identities/lccn-no2008-76892/ (accessed on 15 April 2022)
4	Max Planck Institute for Psycholinguistics	1980	Germany; The Netherlands	A psycholinguistics research institute.	https://www.mpi.nl/ (accessed on 15 April 2022)

**Table 2 children-09-01471-t002:** Bibliometric and Scientometric Indicators for Psycholinguistics Research Adopted from [52].

Element	Definition/Specification/Retrieved Data	Database/Software		
Indicator		Scopus	WOS	Lens
Bibliometric				
Year	Production size by year	√	√	√
Country	Top countries publishing in the field	√	√	√
University	Top universities, research centres, etc.	√	√	√
Source	Top journals, book series, etc.	√	√	√
Publisher	Top publishers	Χ	√	√
Subject area	Top fields associated with the field	√	√	√
Author	Top authors publishing in the field	√	√	√
Citation	Top-cited documents	√	√	√
Scientometric		CiteSpace	VOSviewer	
Betweenness centrality	A path between nodes and is achieved when located between two nodes [53].	√	Χ	
Burst detection	Determines the frequency of a certain event in certain period (e.g., the frequent citation of a certain reference during a period of time) [54].	√	Χ	
Co-citation	When two references are cited by a third reference [55]. CiteSpace provides document co-citation network for references and author co-citation network for authors. In VOSviewer, co-citation defined as “the relatedness of items is determined based on the number of times they are cited together” [51] (p. 5). Units of analysis include cited authors, references, or sources.	√	√	
Silhouette	Used in cluster analysis to measure consistency of each cluster with its related nodes [50].	√	Χ	
Sigma	To measure strength of a node in terms of betweenness centrality citation burst [50].	√	Χ	
Clusters	“We can probably eyeball the visualized network and identify some prominent groupings” [50] (p. 23).	√	√	
Citation	“The relatedness of items is determined based on the number of times they cite each other” [51] (p. 5). Units of analysis include documents, sources, authors, organizations, or countries.	√	√	
Keywords	CiteSpace provides co-occurring author keywords and keywords plus. In VOSviewer, co-occurrence analysis is defined as “the relatedness of items is determined based on the number of documents in which they occur together” [51] (p. 5). Units of analysis include author keywords, all keywords, or keywords plus.	√	√	

**Table 3 children-09-01471-t003:** Search Strings for Data Retrieval on Psycholinguistics.

*Scopus* ( TITLE-ABS-KEY ( “psycholinguistic*” ) OR TITLE-ABS-KEY ( “psycholinguistics” ) OR TITLE-ABS-KEY ( “psycholinguistic” ) ) AND ( LIMIT-TO (DOCTYPE, “ar” ) OR LIMIT-TO (DOCTYPE, “cp” ) OR LIMIT-TO (DOCTYPE, “re” ) OR LIMIT-TO (DOCTYPE, “ch” ) OR LIMIT-TO (DOCTYPE, “bk” ) ) Saturday, 11 June 2022, 12,172 document results, 1946–2022
*WOS* “psycholinguistic*” (Title) or “psycholinguistics” (Title) or “psycholinguistics” (Topic) or “psycholinguistic” (Topic) and Articles or Review Articles or Book Chapters or Books or Early Access or Proceedings Papers (Document Types) Saturday, 11 June 2022, 4845 documents, 1985–2022
*Lens* ( Title: ( AND ( psycholinguistics AND ) ) OR ( Abstract: ( AND ( psycholinguistics AND ) ) OR ( Keyword: ( AND ( psycholinguistics AND ) ) OR Field of Study: ( AND ( psycholinguistics AND ) ) ) ) ) OR ( Title: ( AND ( psycholinguistic AND ) ) OR ( Abstract: ( AND ( psycholinguistic AND ) ) OR ( Keyword: ( AND ( psycholinguistic AND ) ) OR Field of Study: ( AND ( psycholinguistic AND ) ) ) ) ) **Filters**: Stemming = Disabled Publication Type = (journal article, unknown, book chapter, book, dissertation, conference proceedings article, conference proceedings, preprint) Saturday, 11 June 2022, Scholarly Works (15,551), 1946–2022

**Table 4 children-09-01471-t004:** Top-Cited Documents of Psycholinguistics Using Citation Reports from *Scopus*, *WOS*, and *Lens*.

No.	Source Title	Citation	Citations by Database		
			*Scopus*	*WOS*	*Lens*
1	A Cognitive Approach to Language Learning	[61]	Χ	Χ	2202
2	A Solution to Plato’s Problem: The Latent Semantic Analysis Theory of Acquisition, Induction, and Representation of Knowledge	[62]	3758	2991	5162
3	A Spreading-Activation Theory of Retrieval in Sentence Production	[63]	2174	Χ	Χ
4	Activation of auditory cortex during silent lipreading	[64]	Χ	683	Χ
5	An introduction to second language acquisition research	[65]	Χ	Χ	1473
6	Becoming syntactic	[66]	Χ	634	Χ
7	Bilingual language production: The neurocognition of language representation and control	[67]	Χ	668	Χ
8	Concreteness, imagery, and meaningfulness values for 925 nouns	[68]	1776	Χ	Χ
9	Cross-Cultural Psychology: Research and Applications	[69]	Χ	Χ	1725
10	Dorsal and ventral streams: a framework for understanding aspects of the functional anatomy of language	[70]	Χ	1285	Χ
11	DRC: A dual route cascaded model of visual word recognition and reading aloud	[71]	2718	Χ	Χ
12	Expectation-based syntactic comprehension	[72]	Χ	792	Χ
13	Introduction to wordnet: An on-line lexical database	[73]	2503	Χ	Χ
14	Moving beyond Kucera and Francis: A critical evaluation of current word frequency norms and the introduction of a new and improved word frequency measure for American English	[74]	Χ	1424	1827
15	MRC psycholinguistic database - machine-usable dictionary, version 2.00	[75]	Χ	729	Χ
16	On Broca, brain, and binding: a new framework	[76]	Χ	882	Χ
17	Perception of the speech code	[77]	2203	Χ	Χ
18	Reading acquisition, developmental dyslexia, and skilled reading across languages: A psycholinguistic grain size theory	[78]	1743	1597	2113
19	Second Language Acquisition: An Introductory Course	[79]	Χ	Χ	1384
20	The Mental representation of grammatical relations	[80]	Χ	Χ	1439
21	The MRC psycholinguistic database	[81]	1799	Χ	2102
22	The Phonological Loop as a Language Learning Device	[82]	1516	Χ	Χ
23	Understanding Normal and Impaired Word Reading: Computational Principles in Quasi-Regular Domains	[83]	2023	Χ	Χ
24	Word association norms, mutual information, and lexicography	[84]	Χ	Χ	3155

**Table 5 children-09-01471-t005:** Summary of the Largest Psycholinguistics Clusters.

Cluster ID	Size	Silhouette	Label (LSI)	Label (LLR)	Label (MI)	Average Year
*Scopus*						
0	204	0.914	Individual difference	Affective norm (653.29, 1.0 × 10^−4^)	Familiarity account (3.15)	2013
1	162	0.943	Sentence processing	Broca’s area (698.84, 1.0 × 10^−4^)	Refractory effect (0.56)	2002
2	149	0.934	Eye movement	Bilingual language control (820.59, 1.0 × 10^−4^)	Familiarity account (1.43)	2015
3	145	0.874	Relative clauses	Relative clauses (1113.84, 1.0 × 10^−4^)	Familiarity account (1.31)	2007
*WOS*						
0	131	0.896	Formulaic language	Formulaic language (345.43, 1.0 × 10^−4^)	Persistence (1.76)	2008
1	118	0.894	Spanish word	Affective norm (378.11, 1.0 × 10^−4^)	Sub-lexical effect (1.48)	2014
2	106	0.905	Multiplex lexical network	Multiplex lexical network (581.25, 1.0 × 10^−4^)	Persistence (2.28)	2016
3	91	0.85	Cortical dynamics	Cortical dynamics (295.95, 1.0 × 10^−4^)	Persistence (0.33)	2011
4	91	0.927	Language processing	Language learning (264.85, 1.0 × 10^−4^)	Context effect (0.88)	2014
5	87	0.903	Semantic context integration	Semantic context integration (297.47, 1.0 × 10^−4^)	Neurophysiological correlate (0.19)	2003
6	80	0.967	Acquisition norm	Naming latencies (206.52, 1.0 × 10^−4^)	Persistence (0.14)	2002
7	77	0.936	Cross-language perspective	Cross-language perspective (141.17, 1.0 × 10^−4^)	Aphasic speaker (0.03)	1994

**Table 6 children-09-01471-t006:** Citation Counts for Influential Authors in Psycholinguistics.

WOS			Scopus		
Citation	Reference	Cluster ID	Citation	Reference	Cluster ID
92	Bates [89]	2	166	Barr [90]	0
70	Barr [90]	0	70	Bates [89]	2
41	Baayen [57]	0	54	Baayen [57]	3
41	Kuznetsova [88]	2	45	Kutas [87]	5
38	Brysbaert [91]	1	42	Jaeger [92]	3
31	R Core Team [93]	2	37	Brysbaert [91]	0
29	Pickering [94]	4	34	Levy [72]	3
26	Kuperberg [95]	4	28	Kleinschmidt [96]	2
24	Warriner [97]	1	27	Warriner [97]	0
24	van Heuven [98]	1	26	Kuperman [99]	0

**Table 7 children-09-01471-t007:** Detected Bursts for Top Authors in Psycholinguistics.

WOS			Scopus		
Burst	Reference	Cluster ID	Burst	Reference	Cluster ID
34.51	Bates [89]	2	73.98	Barr [90]	0
27.73	Barr [90]	0	31.63	Bates [89]	2
20.62	Baayen [57]	0	24.53	Baayen [57]	3
20.3	Kuznetsova [88]	2	20.76	Kutas [87]	5
15.02	Brysbaert [91]	1	20.12	Jaeger [92]	3
13.25	Levelt [100]	12	16.41	Brysbaert [91]	0
11.94	Pickering [94]	4	16.27	Levy [72]	3
11.57	Kuperman [99]	1	14.37	Kuperman [99]	0
11.55	R Core Team [93]	2	13.47	Indefrey [101]	1
11.55	Kuperberg [95]	4	13.28	Warriner [97]	0

**Table 8 children-09-01471-t008:** Central Authors in Psycholinguistics Research.

WOS			Scopus		
Centrality	Reference	Cluster ID	Centrality	Reference	Cluster ID
54	Bates [89]	2	58	Baayen [57]	3
47	Baayen [57]	0	46	Pickering [94]	5
39	Warriner [97]	1	45	Kuperman [104]	0
38	Alonso [103]	1	41	Jaeger [92]	3
35	Bonin [60]	6	41	Brysbaert [91]	0
35	Abutalebi [67]	8	40	Barr [90]	0
34	Friederici [59]	5	40	Staub [105]	2
33	Kuperman [104]	1	38	Bates [89]	2
32	R Core Team [93]	2	37	Levy [72]	3
32	Baayen [57]	0	37	Warriner [97]	0

**Table 9 children-09-01471-t009:** Sigma Values for Authors in Psycholinguistics with Potential Growth.

WOS			Scopus		
Sigma	Reference	Cluster ID	Sigma	Reference	Cluster ID
0	Bates [89]	2	0	Baayen [57]	3
0	Baayen [57]	0	0	Pickering [94]	5
0	Warriner [97]	1	0	Kuperman [104]	0
0	Alonso [103]	1	0	Jaeger [92]	3
0	Bonin [60]	6	0	Brysbaert [91]	0
0	Abutalebi [67]	8	0	Barr [90]	0
0	Friederici [59]	5	0	Staub [105]	2
0	Kuperman [104]	1	0	Bates [89]	2
0	R Core Team [93]	2	0	Levy [72]	3
0	Baayen [57]	0	0	Warriner [97]	0

## Data Availability

The data presented in this study are available on request from the first author.

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
