# Peer review of "Psycholinguistics: Analysis of Knowledge Domains on Children’s Language Acquisition, Production, Comprehension, and Dissolution"

_children, 2022, doi:10.3390/children9101471_

Round 1

Reviewer 1 Report

This is a review for the manuscript “Psycholinguistics: Analysis of Knowledge Domains on Children’s Language Acquisition, Production, Comprehension, and Dissolution (children-1852302)” submitted to Children. After reading the manuscript, I have the general impression that the manuscript has large room for improvement. First, the scope might not meet the one set forth by Children. Second, the scope of the study is not clearly defined. Third, the Discussion section does not provide insightful discussion. I provide the detailed comments below for the authors’ references.

1.        Abstract: “There is … of the texts.” → This part might be too long. The abstract is expected to be concise.

2.        The purpose and the scope of the current study is not clear and might confuse readers. For instance, to sum up the definitions/scopes of psycholinguistics, the authors state that “psycholinguists study the psychological processes involved in the language usage, including language understanding, language production, and first and second language acquisition [25].”. However, in Table 1 where psycholinguistics journals, associations and research centers are listed, the scope becomes very narrow. There are many journals, associations and research centers focusing on first and second language acquisition. However, those journals, associations and research centers are not listed here. Please be reminded that “first and second language acquisition” are included in the text I quoted above. Another example is from the section “purpose of the present study”. The authors state that “In this study, we examine the development of the field of psycholinguistics and the evidence regarding how children acquire, learn, process, comprehend, and produce language.”. It seems that the scope of “psycholinguistics” as well as the focus of the current study is about children. That is, it implies that studies pertaining to how adult learn, process, comprehend and produce language are not within the scope. However, later in the same paragraph, the authors state that “In addition, it combines bibliometric and scientometric indicators to analyze state-of-the-art psycholinguistic scholarship.”. Therefore, do the authors refer to psycholinguistic studies centering children as “state-of-the-art psycholinguistic scholarship”? In short, the authors must first define what “psycholinguistics” is. After that, the authors must clearly set up the scope of the current study.

3.        Why are Scopus, WOS and Lens selected as the databases? Why aren’t other databases included? Please note that the information provided on p. 9 does not properly address this issue. In fact, do that authors imply that “Lens” does not set up an inclusion criterion? Also, why don’t the authors use Google Scholar as one of the database when it could include even more data than Lens?

4.        The authors are suggested to present the methods of the data collection/analysis process in the section “Methods”. In the current edition, the authors “explains” why and how bibliometrics and scientometrics are used in the current study.

5.        Why do the authors include VOSviewer as one of the tools as the other tool CiteSpace could provide all the information the authors wish to include in the study?

6.        A psycholinguistic study might not include the word “psycholinguistic(s)” in the article. If the purpose of the current study is to “examine bibliometric and scientometric indicators to assess the current state of psycholinguistics research”, why do the authors include one key word only? For instance, according to the definition of psycholinguistics provided by Warren (2012) (c.f., p. 5 in the manuscript), “storage of spoken and written language” is in the scope of “psycholinguistics”. However, most of the articles published in Journal of Memory and Language do not include the term “psycholinguistics”. Nevertheless, most, if not all, of the articles in the journal are psycholinguistics in nature.

7.        The results must be completely modified once the scope has been clearly identified and the keywords for the search are revised.

8.        Several paragraphs in “Discussion” are simply listing the results without further discussion.

9.        The paragraph in the section “theoretical implications” is in fact about “methodological implications”. In addition, the information is available even before the current research is conducted.

Reviewer 2 Report

I suggest improving the discussion to deepen the analysis of the results. 

Authors should cite recent studies such as Guo (2022), another bibliometric study. 

Reviewer 3 Report

 This is an important article that examined the current state of research on psycholinguistics by analyzing publications in three critical databases. It could serve as a reference article at the undergraduate level to learn about the basis of psycholinguistics to the graduate level to investigate where this topic is published and the main contributors to this field.

There are minor aspects to review before publishing the article:

Writing style:

Table 1 needs some capitalization and punctuation on page 7. It’s difficult to identify if it’s only one big paragraph stating the scope of all the journals or if each journal has its own scope. Even if it is a general scope, there is no capital letter after the few periods in this text.

Bibliometric indicators

The information in this section would be more interesting in a table

 Graphs

Why graph C in the figures shows the opposite pattern (highest at the top)?  It would be better to uniform all the graphs.

 Category names in graphs are not uniform, some are all capital letters, and some others are not.

Figure 9 is impossible to read and identify the text (if that’s the authors’ intention). Maybe it’s not the intention, but then could the authors explain more about the graph? Like you did for figure 10 (each color represents a different….)

 Tables

The title in Table 4 needs adjustment. It seems to be “Citations base Scopus” for the 4th  column. Center "Citations by database" so it displays better

Some typos in 1772, Yet they

 Does this sentence need the author and not only the citation? This science according to [17] is obviously linked

Figure 2b WOS, says “documets”

Round 2

Reviewer 1 Report

I am afraid that the results could not meet the purpose set by the authors even after the revision.

1.          Here are some facts. First, the authors claim that “It should be noted that our research in this study is restricted to articles with the term "psycholinguistic*" in their title, abstract, author keywords, and topic. Examining the use of "psycholinguistics" or "psycholinguistic" concepts is the raison d'être of this exclusion.” The author also included the following research questions: 1) What is the size of psycholinguistics' knowledge production over the past seventy-six years as measured by year, region, institutions, journal, publisher, research area, author, and cited documents? 2) Who are the most influential and central authors in the field of psycholinguistics? 3) What are the most sought-after terms and keywords in psycholinguistics? 4) Which patterns are the most investigated and studied in psycholinguistics?

l   However, the size of psycholinguistic knowledge production over the past 76 years could not be measured (by year, region, institutions, journal, publisher, research area, author, and cited documents) when only the key words “psycholinguistics” or “psycholinguistic” are used. Using the words “psycholinguistic” and “psycholinguistics” could not represent, or even approach to represent, the “psycholinguistic knowledge production”.

l   However, the most influential and central authors in the field of psycholinguistics could note be revealed when only the key words “psycholinguistics” or “psycholinguistic” are used.

l   However, the most investigated and studied patterns in psycholinguistics could not be understood when only the key words “psycholinguistics” or “psycholinguistic” are used.

In short, by simply using the key words "psycholinguistics" or "psycholinguistic", the study becomes a corpus study focusing on how the words "psycholinguistics" and "psycholinguistic" are used within a certain period of time. The results failed to reflect or approach to the answers the authors intend to explore.

2.          The authors claim that “This means that, while we are aware that numerous other topics and themes (e.g., first language acquisition, second language acquisition, language learning, child language, etc.) are within the scope of psycholinguistic research, we avoided making our resarch lengthy and instead focused on the use and development of the two concepts listed above.”

l   I find it hard to understand why the authors choose not to include more themes. I check the “Instructions for Authors” section in Children, the journal clearly states that “Children has no restrictions on the length of manuscripts, provided that the text is concise and comprehensive.”. The authors are suggested to utilize this advantage and provide a comprehensive and meaningful study as the journal does not set an arbitrary word limits.

l   Unlike what the authors claim, the current study does not focus “on the use and development of the two concepts, but only two words (i.e., psycholinguistic and psycholinguistics”).

l   If the authors truly think that the length of the manuscript would be one issue, the authors are suggested to delete some of the bibliometrics in the study. Making sure that the study meets the scope of “psycholinguistics” should prioritize other considerations.

l   The title of the study is “Psycholinguistics: Analysis of Knowledge Domains on Children’s Language Acquisition, Production, Comprehension, and Dissolution”, but in fact, the authors exclude other keywords in the study. That is, the results also fail to meet the direction set forth in the title of the study.

l   The word “resarch” should be “research”.

I would deeply appreciate it if the authors could think the suggestions I provided above again.

Author Response

We sincerely apologise for leaving such a negative impression on our paper. We have made every effort to address your feedback as well as that of reviewers 2 and 3. Our main goal is to produce a high-calibre manuscript, and we appreciate all of your feedback.